# Becoming a Socially Responsive Co-Learner: Primary School Pupils' Practices of Face-to-Face Promotive Interaction in Cooperative Learning Groups

Selma Dzemidzic Kristiansen 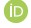

Department of Educational Science, The Faculty of Humanities, Sports and Educational Science, University of South-Eastern Norway, 3045 Drammen, Norway; selma.dzemidzic.kristiansen@usn.no

**Abstract:** Promoting pupils' face-to-face promotive interaction (FtFPI) is crucial for effective cooperative learning (CL) in group work. This article provides insight into interpersonal behaviour and supportive communication as two important aspects of FtFPI. Sixteen pupils 9–10 years of age were videotaped in four structured mixed-ability groups during CL sessions at two primary schools in post-war Bosnia and Herzegovina (BiH). The features of FtFPI that pupils use for peer support in small CL groups and on interfering factors that pupils encounter during FtFPI were analysed using a thematic hybrid approach. The study found that pupils used verbal and non-verbal features for co-learners' responsive actions during FtFPI. However, the findings also revealed some factors that interfere with the pupils' FtFPI, such as having insufficient knowledge and personal skills about peer attention, encouragement and praising. The study recommends that future studies should implement the intervention necessary to foster both teachers' and pupils' understanding and functional knowledge of FtFPI for successful small CL groups.

**Keywords:** face-to-face promotive interaction; cooperative learning; cooperative practice; peer support





## 1. Introduction

Relationships are a fundamental part of successful group work, while supportive interactions are essential for the promotion of learning [1]. The ability of pupils to provide mutual support helps co-learners to make progress towards their joint achievement in small learning groups [2]. Social competencies and the ability to create and maintain effective peer relationships enhance such personal skills as engagement, communication and prosocial behaviour, in other words, skills that are needed if individuals are to be able to connect with others and support each other's academic success [3,4].

This article focuses on pupils' face-to-face promotive interaction (FtFPI) as a type of social interaction that refers to ways individuals encourage and facilitate each other's efforts, thus leading to successful cooperative learning (CL) [5]. However, previous studies have shown that when working in heterogeneous groups, pupils do not spontaneously engage in activities that enhance their learning or necessarily support each other in mastering their learning tasks [6]. In fact, Baines, Blatchford and Webster [7] found that in most primary schools, pupils' group work lacks supportive features.

If group members lack cooperative skills when it comes to co-learners providing and receiving help, they will not work productively in groups [8,9]. Moreover, when pupils choose to avoid or blame rather than engage with another group member, they are showing their inability to build their co-learners' social competence [10]. Hence, more observational studies on CL practices are needed in classrooms worldwide if the increasingly diverse pupil population is to thrive [11]. It is important to focus on the lower grades of primary education due to their high exposure to social, economic and educational disadvantages and the earlier development of their capacity to successfully cooperate [11–13].

CL is widely recognised as a pedagogical practice that promotes small group learning and socialisation [14,15] and leads to positive social interaction and achievement among pupils across different subject areas, where they provide mutual support, share resources and celebrate joint success [5,16]. Bearing this in mind, CL is of interest in the post-war educational reform aiming for a more child-centred pedagogical practice in Bosnia and Herzegovina (BiH) [17]. However, when practising a new teaching method such as CL, pupils and teachers encounter challenges associated with their interpersonal behaviour and supportive communication during cooperative activities [18,19].

CL and FtFPI may be seen as socio-cultural resources for human interaction when learning activities involve supporting others [20]. Employing "mediational means", such as socio-pedagogical tools and language, shapes the pupils' approaches to promotive interactions [21]. In CL, promotive interdependence is a vital component where students engage in promotive interactions by helping each other through support, help and encouragement, and this helps determines pupils' learning outcomes [2]. Thus, following social interdependency theory [22], three interactional dimensions maximize peer promotive interaction success: (a) Substitutability (e.g., the actions of one person substitute for the actions of another), (b) cathexis (e.g., the investment of psychological energy in events outside of oneself), and (c) inducibility (e.g., openness to influence). However, due to the complex relationships associated with challenges and different features of peer support, pupils' FtFPI does not always guarantee that the desired results are achieved due to the problematic practice of CL [23].

The article's point of departure is linked to the theoretical concept of FtFPI. The focus is on pupils' interpersonal behaviours and supportive communication that might contribute to pupils' active engagement as responsive co-learners in small CL group work [1,24]. Thus, the aim of the present study is to understand and discuss how pupils practice FtFPI in small CL groups by investigating pupils' supportive and interfering actions. These actions shape both pupils' openness and responsiveness to others for shared social and academic gains [3,18,19,23]. Specifically, the study attempts to answer the following research questions:

(a)　　Which features of FtFPI do pupils use for peer support in small CL groups?
(b)　　Which interfering factors do pupils encounter during FtFPI in small CL groups?

To address the research questions, the study here focuses on two aspects of FtFPI that have the potential to increase the chances of pupils succeeding in CL: (1) Interpersonal behaviour and (2) supportive communication [24]. Pupils' interpersonal behaviour refers to two dimensions: (a) Recognising that peers need help and (b) willingness to help. Supportive communication consists of interrelated dimensions: (a) Paying attention (b) encouraging peers and (c) peer praising [25]. This study does not only investigate whether pupils encourage, praise and pay attention to each other within group work, but also analyses the ways in which pupils do this.

*Previous Research on Forming and Functioning Aspects of FtFPI*

Promotive interaction is a core element if pupils, exceptionally high-risk pupils and those with individual need, are to benefit from the opportunities CL provides [15]. Pupils are more likely to facilitate each other's learning in mixed-ability groups (high, medium low ability) and gender-balanced compositions [14,26]. However, pupils' behaviour during group work and their joint attention can vary considerably from one group to the next [27,28]. Having skills to communicate effectively through listening, explaining and sharing ideas enables pupils to have more cooperative behaviour [26]. Nevertheless, effective group work also depends on pupils' socioemotional group ethos, taking into account group maintenance and group blocking [13]. Moreover, pupils need to develop prosocial behaviours, such as promoting and seeking help [8] to become responsive co-learners. For pupils' actions to be promotive in CL groups, all the group members must be aware of their own active role in their interaction and be aware of the needs of others [29]. Moreover, pupils' self-confidence may affect their behaviour [30].

Researchers have underlined the need to prepare pupils for promotive interactions [6,31]. A recent review study on aspects of FtFPI has pointed out the importance of preparing co-learners for each aspect of FtFPI [24]. Moreover, pupils need to learn about a variety of FtFPI aspects in line with their forming and functioning dimensions in practice. The term forming dimension refers to the organisation of the group and the establishment of minimum norms for appropriate cooperative behaviour [14]. For this reason, the key roles social interdependence and a joint task play in establishing a group structure that motivates group members to engage in FtFPI and actively support each other's learning [32,33]. Moreover, the collaborative more open-ended task is often suggested as effective in facilitating FtFPI [14]. Accordingly, group members as interdependent co-learners in a reciprocal fashion contribute and exchange resources with others before completing the task [14,23].

Functioning dimensions are needed to manage the groups' activities in completing a task and in maintaining effective working relationships among pupils (e.g., asking for help, expressing support) [2]. Moreover, affective factors such as pupils' socio-emotional experiences may influence CL group work and originate from each group member's perceptions of his/her peers during the interactions [34]. In addition, a group member's personality traits such as self-consciousness and self-monitoring may also contribute to the role of learner-facilitator during FtFPI [35].

Previous research has increased our understanding of specific aspects associated with pupils' FtFPI, for example, seeking and providing help [9]. Pupils' responsiveness to others [36] and their willingness to seek and give help [37] have been recognised as initial dimensions of interpersonal behaviour in FtFPI. Webb and Mastergeorge [9] highlight that high-quality help is only useful to the receiver when it is sufficiently elaborated on, corrected on time and linked to the need for help. However, the most accurate predictor of positive support is whether the receiver of the help makes use of it [38].

To promote the pupils' ability to provide mutual support in co-learning tasks [16], the verbal and nonverbal behaviour that is part of supportive communication requires active listening, paying attention and encouraging and praising others [25]. Moreover, using supportive communication that can serve as a peer model that others can and should imitate is a way of helping pupils to achieve successful FtFPI [39]. A supportive peer model refers to behaviour that occurs when pupils observe other pupils' actions and then imitate them as an incentive to help others [40]. However, the teacher's role in modelling helping behaviours is crucial for effective pupils' help-related conduct during small CL group work [9].

Moreover, the teacher's role includes the structuring of group work for cooperation and status relations in interaction [41]. Following up on the social norms for interaction, teacher's monitoring and intervening occurs in the group work when needed [9,31,42]. While balancing pupil status can play a critical role in making cooperation in small groups successful, teachers must create a group-worthy task that requires each member's contribution and the help group members offer one another [14,43].

## 2. Materials and Methods

The present descriptive case study [44] took place in two purposefully selected primary schools [45]. Qualitative video data were collected on interpersonal behaviour and supportive communication that enabled the researcher "to dig into" the pupils' FtFPI as a complex practice, thus allowing her to look at a particular FtFPI situation several times [46].

### 2.1. Context and Participants

In post-war BiH, an education reform introduced a child-centred educational process based on participatory, active and cooperative methods aiming to harmonise the quality of teaching and learning practices with contemporary European teaching and learning models [17]. However, the educational system is still highly complex and fragmented thus that the problem of divisions and discriminatory behaviour limiting human cooperation

on educational progress continues [47]. Moreover, systematic measurements of the quality of the education and scientifically based data on pupils' learning are lacking [48].

The case in this study presents School A and School B, 2 institutions that have implemented the reform efforts by moving from teacher-led to child-centred pedagogical practices. The schools were located in a socioeconomically less-privileged area of Sarajevo where the pupils' families were dealing with such post-war consequences as trauma, migration from other parts of BiH, low-income, one-parent family and a minority group of Roma people. Thus, while dealing with adversity and diversity and coupling this with the power of cooperation, these schools were focusing on CL activities thus they could facilitate their pupils' learning process. Two classrooms, one from each school, were selected based on teachers' willingness to participate, and their pupils were involved in CL experiences 2 to 3 times a week across various school subjects.

Sixteen pupils were selected from a larger sample (N = 192). The selected pupils were engaged earlier in the previous study that explored a deeper understanding of their perceptions about key aspects of FtFPI by conducting a face-to-face interview [18]. Accordingly, the pupils' perspectives on the FtFPI [18] and the present video observations may provide a complete picture of FtFPI' situations in small CL groups [49]. Using the pupils' grades in the class' protocols, the teachers chose a sample of 16 pupils 9–10 years of age (8 boys and 8 girls), as the power of mixed academic levels or mixed social status supports learning among peers [14]. The selected pupils have been in the same class from Year 1, and the same pupils were invited to participate in the present study. The teachers and pupil's parents gave written consent for their own and their child's participation in the study. The participants had no additional preparation relating to FtFPI for CL other than the child-centred methodology. The teachers' instructions in both classrooms about a joint task and cooperation between pupils before the group sessions required that everyone cooperated, everyone listened to each other, shared their knowledge and helped one another [38]. Sometimes, pupils themselves were asked to remind group mates about these rules along with helping behaviour.

### 2.2. Data Collection

Two gender-balanced groups in Year 4 in both schools (N = 4) were videotaped throughout group sessions across the subjects Mathematics, Bosnian language and Natural Science in the spring of 2019. Each group consisted of 1 high, 2 medium and 1 low level achiever (N = 16) (see Figure 1). The intention behind the video recording of the groups was to examine in detail the current pupils' practices in relation to FtFPI, such as promotive actions and actual supportive or interfering dialogues [50].

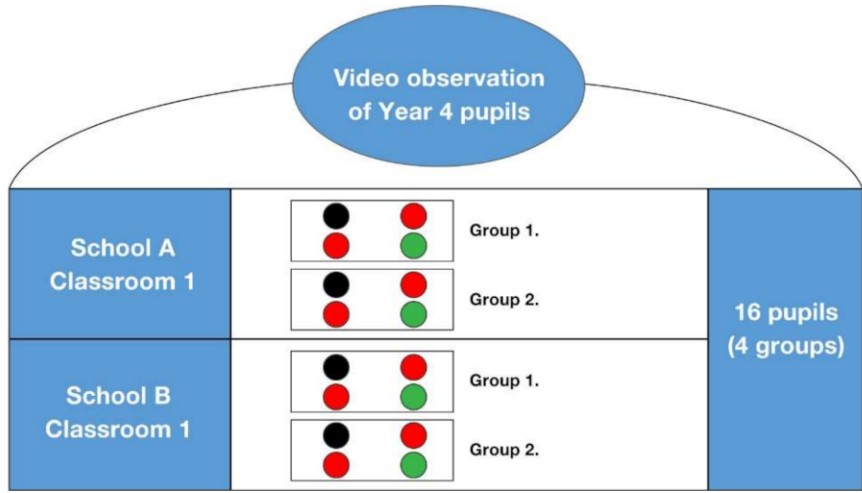

**Figure 1.** An overview of the setting for the video data collection.

The study had a total of 11 h and 27 min of recorded material, including 2 to 3 sessions a day in the same groups. The length of the sessions varied between 15 and 30 min depending on the assignment given. The collaborative assignment, which was very different in nature and content, was designed for CL purpose with an open-ended question and a strong narrative structure. The teachers planned the authentic assignments to engage pupils in joint productive activity (see Appendix A). Using dialogical and analytical skills, pupils worked together toward a common goal (e.g., creating mind maps, making a report/common argument for class debate or solving mathematical problems). All mathematical tasks were adjusted from the regular mathematical curriculum and were embedded in contexts exercising together. Sometimes pupils thought or wrote individually, and later they discussed the solution for the problem as a group. For each transcript of videotaped sessions across the school subjects, the researcher developed codes including the date, the school, the group, and the session number, e.g., SA-G1-S1 (School A, group 1, and session 1). Each code interpreted the session schedule, including the school subject, joint task, and main purpose.

Two cameras (Zoom Q2n Handy video recorders) were placed on a tripod and angled on the pupils' group work, including two dictaphones (H1n Handy Recorders), each in Group 1 and Group 2 (see Figure 2).

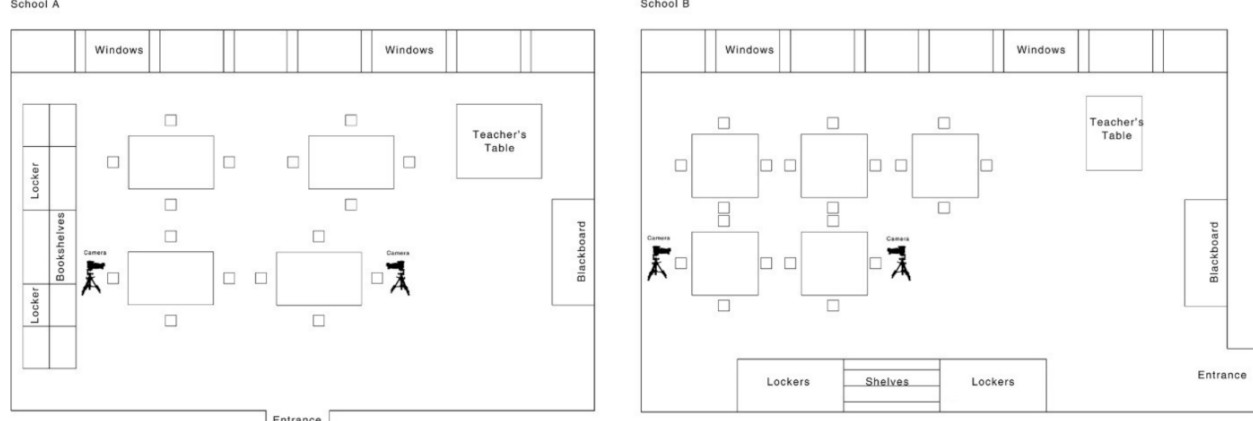

**Figure 2.** Classroom maps of recorded group work.

Moreover, the researcher recorded "off-camera" contexts by using the observational protocol [45]. The questions that guided this researcher's observations were "What did the pupils do when they left their groups? Did they ask for help from their teacher or peers outside their group, and what happened later?" These notes were useful for understanding and capturing the context within which the pupils interacted and were later incorporated into the videotaped transcripts [51].

The "appropriateness" of this research process and data were addressed in the internal and external validity check. The data material were collected in the authentic setting of primary-school classrooms, while the videos provided the opportunity to review the pupils' group and individual actions over and over for their accuracy [51].

Prior to the data collection process, the researcher addressed practical and ethical issues, such as acquiring informed consent, gaining trust and avoiding misunderstandings relating to the pupils' participation during the entire research process, as well as storing, organising, analysing and presenting the videotaped material [52].

### 2.3. Data Analysis

Thematic analysis employing a hybrid approach of deductive and inductive reasoning was utilised with the pre-defined FtFPI categories that were both a precursor to and an outcome of the data analysis [50,53]. The researcher transcribed and coded the video data. This process included searching for and identifying common features that extended across

FtFPI fragments divided into interpersonal behaviour and supportive communication in CL groups [44,54] (see Appendix B). Through a bottom-up, inductive, data-driven process, emerging themes from the participants' activities were refined, organized and added to FtFPI's categories [55,56]. The analysis was viewed as ongoing and iterative, requiring the researcher to constantly question the transcriptions by writing reflection notes while viewing the videos [53].

First, the researcher produced rough transcriptions without specific marking details, such as gestures [55], typing them electronically in the participants' mother tongue, Bosnian. The choice of manual analysis of qualitative data allowed the researcher to read the data, use colour-coding to mark parts of the text and divide them into segments according to the pre-defined FtFPI's categories. Operating within a small database, with fewer than 300 pages of transcripts, the researcher could efficiently sort and organize text sentences into file folders by having a hands-on feel close to the data [45].

The second step of the analysis consisted of careful reading and rereading the transcripts while viewing the videos to obtain a general sense of the FtFPI situations. Next, the researcher refined the transcriptions as specific "key video-clips" relating to FtFPI by adding the multimodal features of the data for the microanalysis [57] (see Appendix C). A Bosnian primary-school English teacher served as an external auditor and collaborative partner in post-recording phases [45]. She reviewed concurrence between the video clips and the transcripts and later translated the transcriptions into English. We also shared responsibility for the data analysis to review the findings and discuss the links between the actual empirical data and the multimodal features added to justify the interpretations [46]. This member-checking process [45] was used to reveal any biases and carefully support the basis for the data interpretation.

A unit of analysis was the video excerpts [55], where the FtFPI based on pupils' activities was identifiable and defined by the FtFPI sub-categories [18] (see Appendix B). Seeking to understand the pupils' interpersonal behaviour and supportive communication, the purpose of this phase of analyses was to extract the supportive and interfering features associated with both recognition and willingness to help each other, and encouragement, praising and paying attention to each other [50]. In addition, the analytical strategy focused on verbal and non-verbal features, using line numbers to help identify the location of these specific segments [58]. Although special attention was paid to pupils' lines, teachers' lines were also included in the analysis since teachers played a key role in FtFPI's development among pupils [9,31,41,43].

The microanalysis started by focusing on the groupmates' engagement in helping and supporting situations with peers during a joint task. Then, the researcher examined whether groupmates reacted by using verbal or nonverbal features; how and when their peer needed some help. The supportive features and interfering factors, and the words and gestures that pupils used to support learning together were examined. The microanalysis also paid attention to teacher's activities identifiable in the recorded groups to obtain an insight into teacher's engagement in supportive relationships.

## 3. Results

The excerpts (N = 10) below have been chosen for detailed analysis of FtFPI in small CL groups. Categorising FtFPI (see Appendix B) into sub-categories of (1) interpersonal behaviour and (2) supportive communication, the findings report (a) features of FtFPI that pupils used to support their co-learning, (b) interfering factors identified during FtFPI and presented in each FtFPI sub-category.

### 3.1. Interpersonal Behaviour

The analysis of pupils' interpersonal behaviour refers to such interrelated dimensions as recognition and willingness to receive and provide help as a response to peers' needs during FtFPI.

Peers' Recognition and Willingness to Help

Excerpts 1 and 1.1 from the same group session are examples of how pupils recognised certain cues indicating a peer's need for help. However, in the first example, the pupil who needed help rejected the offered assistance, which then impeded the group work at that particular moment. One group member then intervened to help the group members to continue their work.

**Excerpt 1 (Each pupil in the group has been numbered in the following way: HLAg pupil, girl with high-level achievement, MLAb pupil, boy with mid-level achievement, LLAb pupil, boy with low-level achievement and so forth).**

1. HLAg: "We all have the right result for this one."
2. MLAg: "Stop!" (.) "for (D.)" (LLAb)
[...]
4. LLAb: "I'm just about to. (Hhhhh)" (rests his head on his elbow)
5. MLAg: "Do you know how to solve it at all?"
6. LLAb: "I know"
7. MLAg (moves closer to LLAb, leans over his notebook)
[...]
11. MLAg: "Well, just tell me where you got that two from!" (She gets up a little from her chair and leans even more over to see what LLAb is writing): "You should have done that in the beginning."
12. LLAb (starts erasing).
13. MLAg: "Well, just tell me (..) where did you get that two from?"
14. LLAb (takes the notebook from the desk and closes it)
15. HLAg (looks at LLAb with her serious face): "The teacher said that our notebooks should not be closed."
16. LLAb (puts his notebook on the desk again and opens it)

MLAg stopped the group activity by recognising that LLAb was still working (1) and became aware that LLAb needed help by posing him questions relating to the task (5, 11). While LLAb claimed that he understood the task (4, 6), his audible exhaled sigh while positioning his head on his elbow (4) and erasing something (12) showed the opposite. MLAg was persistent in offering him help (13) that in turn influenced LLAb to close his notebook (14). To get LLAb back to work, HLAg used the authority of her gaze, thus invoking the authority of the teacher (15).

**Excerpt 1.1**

1. LLAb (looks at MLAg's notebook): "I don't get this at all."
2. HLAg: "So, 8 divided by 4 (..) you see here how much that is."
3. MLAg (moves closer to them): "Write two!"
4. HLAg: "Because 4 times two can be eight (.) right?"
5. LLAb: "Yes"

HLAg reacted at the right time (2), after LLAb stated his confusion and by looking in MLAg's notebook (1). HLAg's proximity to LLAb (2) and MLAg's body movement closer towards LLAb and HLAg (3) indicated their openness and HLAg's willingness to help. However, it can be discussed whether MLAg's answer and HLAg's explanation were the proper way of providing help in this situation (3, 4).

Below is an illustration of a "peer help recogniser" who could not provide help but indicated who might be able to help.

**Excerpt 2.**

| |
|---|
| 1. LLAb (is leaning on his elbow while holding his forehead and looks at the worksheet) |
| 2. MLAb: "Ask (N.)" (points at HLAg) |
| 3. LLAb (calls HLAg): "(N.)" |
| [. . . ] |
| 6. HLAg (looks at MLAb): "And why don't you help?" |
| 7. MLAb: "I'm not sure myself (.) It's better to ask you." |

When MLAb realised that he could not help (7) while the body language of LLAb indicated his need for help (1), MLAb gave LLAb the incentive to ask for help (2). However, the potential helper, HLAg, did not seem to be willing to help (6).

Excerpts 3 and 3.1 below illustrate the groupmates' non-response to solicited help (3) and non- willingness to help because the "helping points," previously assigned to pupils in need (3.1) within the same group session, had been used up.

**Excerpt 3.**

| |
|---|
| 1. LLAg (coughs a bit): "Here it is (.hhh)" |
| 2. MLAg (looks at the worksheet of LLAg and slightly frowns) |
| 3. LLAg (looks away from MLAg's face and onto her worksheet) |
| 4. MLAg (raises her eyebrows a couple of times and sticks out her tongue a bit) |
| 5. LLAg (quietly): "I want someone to help me" (groupmates are occupied with work) ( . . . ) |
| 6. LLAg (raises her hand) ( . . . ) (rises from her chair, looks at HLAb): "I have to tell the teacher something." (leaves the group) |

After slight coughs and an audible inhaled sigh made by LLAg (1), MLAg recognised these non-verbal cues as an invitation to give some kind of help to LLAg (2). However, MLAg did not offer any task-related help other than her facial expression signalling that something was wrong in LLAg's work (4). That, in turn, triggered LLAg to ask for help explicitly (5, 6). However, the group did not react, and LLAg left the group to seek external help (6).

As the group work continues, the groupmates more clearly stated that they could not help anymore because LLAg had used up all her "helping points" (1).

**Excerpt 3.1.**

| |
|---|
| 1. MLAb (looks at LLAg): "So, we can't help you anymore (.) you've spent all three points, you're in the hole" |
| [. . . ] |
| 3. Teacher: "Did anyone make a lot of mistakes?" |
| 4. HLAb (loud): "(V.) (LLAg) used all three points." |
| [. . . ] |
| 7. Teacher (approaches the group where LLAg is sitting): "That's not, that's not much (..) come on." |

HLAb's confirmation that LLAb had exhausted his opportunities for help (4) and the teacher's encouragement concerning LLAb's mistakes (7) show the possible detrimental consequences of pupils' willingness to help and reactions to help when applying "helping points."

Below, the same group, but in another session, needed their teacher's intervention to activate the pupils' willingness to give peer help.

**Excerpt 4.**

| |
|---|
| 1. Teacher (looking at the LLAg's notebook): "What task did you come up with?" |
| 2. LLAg: "We-e-e-ll." |
| 3. Teacher: "Which one was yours (V.)?" (but teacher looks at HLAb) |
| 4. HLAb: "She needs to do this one" (..) "111" |
| 5. Teacher (turns the handout to LLAg): "Come on (..) You have numbers 111 and 8." |
| 6. MLAb and MLAg follow while the teacher helps LLAg |
| 7. LLAg (looks at the teacher): "I don't know what I should end up to with ( . . . ) Can I get some help?" |
| 8. Teacher: "What can you suggest to her?" (looks at HLAb and MLAb in turn) |
| 9. MLAb (looks at the teacher and LLAg in turn): "Write this . . . " ( . . . ) (looks up) |
| 10. Teacher: "How many boxes ( . . . ) and the number of pieces is . . . ?" |
| 11. HLAb: "Write it like this." |

As the teacher realised that LLAg had not finished her task (2), the teacher's gaze activated HLAb's willingness to help by reminding them about LLAg's task (3). Moreover, the teacher's explicit verbal invitation along with her gaze directed on HLAb and MLAb (8) initiated their willingness to help. However, MLAb did not seem to have a readiness to help, which he showed by pausing and looking up (9). Thus, HLAb only offered help (11) after the teacher posed the task-related question (10).

*3.2. Supportive Communication*

This section presents the analysis of groupmates' encouragement, praise and attention as three interrelated dimensions illustrating pupils' verbal and non-verbal features used during (non)- supportive communication.

3.2.1. Paying attention and praising

Excerpts 5 and 6 present the same group, but in two different sessions illustrating helping situations. In particular, the group leader HLAg pays attention to all members of the group, including all who are in the task-related conversation, and praises their efforts while simultaneously offering peer assistance.

**Excerpt 5.**

| |
|---|
| 1. HLAg: "So, 23 times 32 ( . . . ) what are we to write and where?" |
| [. . . ] |
| 4. HLAg (addresses LLAb): "Let me see how you're getting on." |
| 7. LLAb (shows in his notebook) |
| 8. HLAg: "Bravo!" |
| [. . . ] |
| 10. HLAg: "Three, ( . . . ) let me see . . . put this a bit higher(..) a bit h-i-i-i-i-gher." |
| [. . . ] |
| 14. HLAg: "(K.) (LLAb) . . . how much is 2 times 3?" |
| 15. LLAb: "Six." |
| 16. HLAg: "Bravo! (.) And we're to write it below what?" |
| 17. LLAb: "Below 2." |
| [. . . ] |
| 25. HLAg (calls to MLAg): "How much is two times two?" |
| 26. MLAg: "Four." |
| 27. HLAg: "And where are we supposed to write it below?" |
| 28. MLAg: "Below four." |
| 29. HLAg: "Bravo!" |
| 30. HLAg (calls to MLAb): "How much is 3 times 3?" |
| 31. MLAb: "We write nine below four." |
| 32. HLAg: "Bravo!" |

HLAg attracted her groupmates' attention in order to evoke their understanding about the procedure for solving the task (1). She began by helping LLAb (4) and praised his efforts along the way (8, 16). Using a slow dynamic for her voice and by taking short pauses, HLAg supported LLAb's task understanding (10). HLAg paid attention to other groupmates by inviting them to confirm their understanding of the task procedures, which HLAg also commended (25–31).

As the group leader continued to have her full attention on helping LLAb, the class teacher explicitly praised this situation, particularly HLAg's efforts.

**Excerpt 6.**

| |
|---|
| 1. HLAg (focused on LLAb): "How much is 4 divided by 4?" |
| 2. LLAb: "Zero." |
| 3. HLAg (repeats in a slightly different questioning tone): "4 divided by 4?" |
| 4. LLAb: "Two." |
| 5. HLAg: "Four divided by four?" (little slower while looking at him) |
| [. . .] |
| 8. LLAb: "One." |
| 9. HLAg: "Bravo! Because you always need to check how many times 4 can go into 4." |
| 10. Teacher (approaches the group): "How's it going (M.)?" (HLAg) |
| 11. HLAg: "Good . . . Good." |
| 12. Teacher: "Super... Hats off." (pats HLAg on her head) |
| [. . .] |
| 77. Teacher: "Here hats off! Applause for (M.) She works so hard and help." (everyone applauds) |
| [. . .] |
| 92. HLAg: "(K.) (LLAb) please, always tell me if you don't understand a task." |
| "If you think you know (..), don't be ashamed." |
| 93. LLAb (nods) |
| 94. HLAg: "If you make a mistake ( . . . ) it doesn't matter. It's okay!" |

HLAg showed her patience in guiding LLAb to answer properly, repeating the same question, changing her questioning tone and the dynamics and timbre of her voice (1–5). Beyond a task-related explanation (9), HLAg encouraged LLAb's insecure behaviour in group work (92, 94). Their teacher was aware of this and praised this helping situation with the word "super" and the metaphor "hats off" (10, 12), initiating the pupils' applause (77).

3.2.2. (Dis)encouragement

The findings in excerpts 7 and 8 reveal that one groupmate's positive or negative attitude can (dis)encourage the further flow of the group work.

**Excerpt 7.**

| |
|---|
| 5. LLAb: "So we only did two tasks." |
| 6. MLAg: "What to do, that's what we have on the desk." |
| 7. MLAb: "Maybe it's not too late. Let's try! Never give up." |
| 8. LLAb (looks at MLAb and smiles): "Let's try" (addresses HLAg) |
| 9. HLAg: "If we put ( . . . ) branch 4 . . . |
| 10. MLAb: "We put 4." (adding cheerfully) ( . . . ) "Never give up!" |

MLAg and LLAb (5-6) expressed their dissatisfaction over what they had done thus far. However, MLAb started to encourage other groupmates to continue (7, 9) by showing his positive energy and using a cheerful voice (10). Ultimately, HLAg began by suggesting how to proceed on the task (9).

On the other hand, one groupmate's negative attitude to the assigned task may discourage the group from starting to work.

**Excerpt 8.**

| |
|---|
| 1. MLAb: "This is the most difficult task that we've got in the group." |
| 2. HLAg: "The teacher thinks (.) we're good pupils ( . . . ) we'll do it easily." |
| 3. MLAb: "No, that's certainly not true." |
| 4. MLAg: "We've got nine more minutes." |
| 5. MLAb: "The minutes go by like this." (snapping of his fingers) |
| 6. Teacher (approaches the group): "Yes, you can do it!" |
| 7. MLAb: "Teacher, why have you given us this task?" (somewhat plaintively) |
| 8. Teacher (smiles): "Let's get down to business." |

MLAb was complaining that their group task was very difficult, but HLAg tried to encourage him by explaining why this had been assigned to them (1, 2). However, MLAb explicitly disagreed with HLAg's explanation (3). Attempting to turn this discouraging atmosphere around, MLAg warned that time was running out (4), but MLAb kept being negative (5). Indeed, in reply to the teacher's encouragement (6), MLAb complained yet again (7).

Excerpts 9 and 10 illustrate how lack of peer attention among groupmates influences pupils' working relationships.

**Excerpt 9.**

| |
|---|
| 1. Teacher: "You've got five minutes." |
| 2. HLAg: "Hurry up!" (frowns and looks at MLAg) |
| [. . . ] |
| 6. HLAg: "We'll never finish this." |
| [. . . ] |
| 10. HLAg: "Look how ugly you're writing . . . Oooh, my God!" |
| 11. MLAb: "Look how her letters are so small." |
| 12. MLAg (angrily pushes the paper away): "Okay! You write." |
| 13. HLAg (returns the sheet of paper with a smile): "You do it." |

HLAg showed her nervousness by rushing MLAg to finish their task (2), remarking negatively about the group's progress (6). Moreover, HLAg's negative comments about MLAg's writing (10) also triggered MLAb to add a negative comment (11). This caused MLAg to stop writing where she angrily pushed the task over to HLAg (12).

Excerpt 10 shows the teacher's intervention after one groupmate has left the group.

**Excerpt 10.**

| |
|---|
| 1. Teacher (approaches the group): "What is (V.) (LLAg) doing?" |
| 2. HLAb: "She wants to draw while we're writing this." |
| 3. LLAg (returns to the group) |
| [. . . ] |
| 5. Teacher (addresses LLAg): "You see, you draw, you're creative! |
| [. . . ] |
| 7. LLAg (addresses her group): "You see (.) teacher claims, I'm creative." |
| 8. HLAb: "We told her that you're drawing (..) we're just supposed to write things down." |
| 9. LLAg: "Then I'm sorry I didn't hear that." |
| 10. HLAb: "You didn't hear us." |

The teacher was fetched by LLAg to intervene in the joint task (1–3). Encouraged, LLAg (5) showed her self-confidence by repeating the teacher's words (7). However, it seemed that failure to pay attention and listen attentively to each other was what led LLAg to leave the group to seek teacher intervention (8–10).

## 4. Discussion

The aim of the study was to investigate which features of interpersonal behaviour and supportive communication of FtFPI the pupils used in small CL groups and which

interfering factors the pupils had to deal with. Thus, the interrelated supportive and interfering dimensions associated with the two mentioned aspects of FtFPI will be discussed to shed light on the research questions. Moreover, the theory of social interdependence [22] provides the framework for the discussion on the FtFPI dimensions.

Building on the knowledge of which didactic and pedagogical support of learning is appropriate for each group learning situation [16], this study has attempted to contribute to research by exploring pedagogical factors in interpersonal behaviour and supportive communication that might be conducive to and constructive in maximising pupils' FtFPI and thus having successful CL group work. Therefore, the present study supports the finding that it is necessary to understand pedagogical tools to have effective social interaction in CL [10].

### 4.1. Recognition and Willingness to Respond to Peer's Needs

The findings point out that the supportive dimensions of interpersonal behaviour among pupils across small CL groups provide certain indicators for recognising peers' need for help and their willingness to respond to it. Some of these are verbalised as explicit requests for help or general statements of confusion such as "I don't get this at all," which has also been found in previous studies [9,18]. The present study also identifies non-verbal signals, for instance, pupils use audible exhaled sighs or slight coughs together with their upper body movement as potential cues for wanting help. Some pupils look into their peers' notebooks, and this may then initiate their groupmate's reactions as a response to a possible need for help. Accordingly, pupils' responsiveness to others and their willingness to seek and give help increase efforts to engage groupmates in FtFPI for successful CL [28]. In most of the excerpts, the peer's need for help is recognised. However, the pupils do not always show a willingness to help for reasons that will be discussed below as interfering factors within FtFPI.

The micro analysis gives insight into how the pupils demonstrate their willingness to help [26] that may lead to better understanding of the peers' implicit needs [36]. For example, Excerpt 1.1 shows how the peer helper and peer receiver create a resource to indicate willingness in the help process through their body postures and proximity [27]. The same excerpt shows that continuity in helping and peer modelling are an incentive for other groupmates to orient themselves towards helping the receiver [39]. However, the quality of task-related help remains questionable. Moreover, the findings suggest that pupils' abilities to recognise the need for help and to be willing to help are crucial aspects for forming and functioning in FtFPI, but they are not sufficient for joint task achievement [2]. Groupmates' knowledge and skills in helping others during FtFPI will be successful if the help giver provides elaborate explanations and monitors the pupils' understanding of the explanations and their ability to apply them [9]. Excerpt 5 demonstrates the above-mentioned approach to help where all the groupmates are included in the supportive process of co-learning. However, many of the excerpts show that the receiver of the help must first be actively included in the FtFPI process. Bearing this in mind, all group members must be self-aware of their active role in FtFPI [29].

The findings reveal three interfering factors that influence pupils' responsiveness and willingness to help. The first is the lack of personal attention invested in FtFPI. On the one hand, either the potential help receiver or the help giver does not show interest, but on the other hand, if the pupil's willingness to help is too intrusive, as in Excerpt 1, the help receiver's behaviour will be affected. Similar to this finding, the pressure from high-ability pupils to complete tasks quickly undermines the participation of the less able [59]. The second factor is that relevant knowledge and skills relating to helping strategies are lacking. Excerpts 2 and 6 show that the pupil's self-confidence and their lack of willingness to help others are related to the lack of a helping strategy. Similarly, Yoruk [30] reported that pupils' self-confidence and self-efficacy affect their cooperative behaviour. Third, two external factors have been identified in the present study that affects pupils' FtFPI: (1) Pupils' dependency on the teacher's intervention to incentivise

and increase the willingness to help, (2) the use of co-called "helping points" may impede FtFPI or decrease the willingness to help, as documented in Excerpt 3.1. Rather than using extrinsic motivation, pupils should have intrinsic motivation to strive for the common good where each pupil sees their own achievement as a possible service to others [5].

According to social-interdependence theory [22], the responsiveness and willingness to succeed in FtFPI for the common good require an understanding of both oneself and others. Moreover, the pupils' inducibility should be a trigger for social and individual mediation in cooperative groups responding to peers' needs and supporting the CL process [19]. Peer support through FtFPI facilitates both social and academic learning, especially for disadvantaged groups where peers play an active role in the induction of new or less able members into a cooperative community [40]. However, the interfering factors presented here, and which concur with the findings in previous studies [18,27] reveal a lack of peer attention and insufficient knowledge of how to help peers to work in small groups that are aiming to be cooperative. For this reason, the teachers' role supports the multiple ability treatment and assigning competence to low-status pupils' cooperative group work in terms of equal access to the group task [14,41]. In Excerpt 10, the teacher's intervention helps the LLAg pupil become self-aware of her creative ability. By doing this, teachers raise the status of low-level pupils by providing more public recognition that everyone has an important ability to contribute to group work by altering the expectations for competence that pupils may hold to each other [41].

### 4.2. Supporting Others through Supportive Communication

To maximise the potential of FtFPI, the interconnected aspects of paying attention, encouraging and praising are crucial for group functioning and managing peer support in CL [18,25]. The findings in the present study indicate that pupils use several pedagogical tools, verbally and non-verbally, to support their groupmates' work. The analysis across the excerpts found that to praise their groupmates, the pupils used the word "Bravo" or applauded, and they would also smile, nod or say "come on" to encourage groupmates. However, there is a need for more than "Bravo" and "Come on" when praising and encouraging others' participation while working together. Pupils' variation in the use of pedagogical tools is necessary to support more connectedness between groupmates, such as making explicit efforts to involve others and getting them to participate [3,60] and prevent discouraging situations from arising in FtFPI. Accordingly, Year four pupils believe that knowing more about how to encourage and praise peers may improve their co-learning, particularly the boys, who lack sufficient knowledge in this area compared to girls [18].

As a positive example of supportive communication, Excerpt 5 demonstrates an inclusive style practised by the group leader, who simultaneously pays attention to a less able pupil and other groupmates. Richmond and Striley [61] argue that the inclusive leader should bring the task-related question to everyone's attention, ask group members for their opinions and encourage their participation. In Excerpt 5, HLAg is an inclusive leader who uses a dynamic voice and timbre by taking short pauses, combined with a facial expression and mindful gaze during FtFPI. While these tools regulate the groupmates' attention, they may also support the LLA pupils' understanding of the task. Moreover, in Excerpt 6, the same pupil, HLAg continuously praises each effort and the answers of an LLA pupil by saying "Bravo." Praising LLA pupils who demonstrate a particular skill and then linking that ability to task requirements reduces the gaps in status in heterogeneous groups [41]. By doing this, HLAg expands her encouragement of LLAb in advising her peer how to be more self-confident during group work. HLAg seems primarily to want the LLAb groupmate to succeed. According to the social interdependence perspective, pupils help each other to learn because they care about the group and its members [15,22].

In turn, the teacher who monitored the FtFPI situation praises HLAg's patience and commitment to the help receiver and initiates the pupils' applause, serving as group praise. Accordingly, the teacher demonstrates guidance on the CL skills of individual pupils

and the group as a whole that support the pupils thus they cooperate effectively [38], in particular stimulating their supportive communication. However, the teachers did not give pupils specific feedback on their cooperative behaviours nor asked them to reflect on how the group behaves concerning FtFPI. The CL literature specifies that teachers need to monitor, support and consolidate the pupils' interaction, including group processing as a tool for reflection for successful group learning [2,62].

Otherwise, the pupils in some of the groups in the present study show a dependency on their teachers' involvement to regulate their FtFPI. Without an appropriate knowledge base, or the ability to organise processes such as FtFPI, pupils are more dependent on their teachers to help them take more control of their learning process [26,40,63]. Excerpts 10 and 4 show situations where the teacher regulates LLAg's involvement in a group task and encourages groupmates to work together as they have insufficient knowledge about FtFPI. These findings concur with other challenges that undermine supportive communication in joint CL activities due to a lack of cooperative skills or not knowing how to provide help and encourage peers [8,18].

When groups lack sufficient strategies for dealing with group maintenance and stalled cooperation, the situation can become very tense and frustrating for all involved [60]. The present study identifies particular factors interfering with or stalling FtFPI that relate to the cathexis of the pupils' positive or negative investment of their own energy in each other's actions that may determine their progress or lack of progress in FtFPI [22]. First, a groupmate's negative speech or gestures relating to the progress of the learning process or assigned group task discourages the working atmosphere among group members, as MLAb demonstrates in excerpts 8 and 9. Conversely, a groupmates' positive attempts support the group work and encourage groupmates to continue, as another MLAb shows in Excerpt 7. Accordingly, a positive group member's personality characteristic may serve as a resource to facilitate socially responsive co-learning during FtFPI [34,35]. However, pupils need to be empowered by personal skills through supportive communication and prosocial inter-personal behaviour to connect with others, avoid interfering factors and sustain FtFPI based on positive interdependence [3,23].

## 5. Conclusions and Recommendations

This study has shown that working together consists of many different aspects of inter-personal behaviour and supportive communication as a key to enabling pupils' support in a highly complex process of FtFPI. Specifically, this study investigated supportive features and interfering factors of FtFPI that shape pupils' openness and responsiveness to others, leading them to be socially (non)responsive co-learners for shared social and academic gains [3,18,19,23]. Research findings reveal that verbal and non-verbal features of FtFPI can be conducive to maximising pupils' recognition and willingness to help, thus leading groupmates might pay more attention to, encourage and praise one another in small CL groups. If pupils have insufficient social skills and lack practical knowledge about FtFPI, supporting one another's needs is not an easy practice, as demonstrated in the present study. Engaging in socially responsive co-learning requires its deeper understanding. The use of supportive socio-pedagogical tools and practical strategies for group maintenance and peer support is particularly needed for pupils to respond to one another's needs towards group success.

Moreover, the teacher's involvement and pupils' background characteristics [11,38] are important dimensions to consider if FtFPI is to lead to successful cooperative groups, where groups can be seen as an arena of personal and collective socially responsive development. In particular, Ferguson-Patrick [11] points to the importance of an engaging and caring environment with social responsibility and concern for others in supporting pupil growth and learning. This study can guide future intervention studies aimed at improving factors that support or impede pupils' group learning, promotive interaction and prosocial behaviours (see also the SPRinG programme of Baines et al., [7]) and Complex Instruction [14,41]. In a particular context aiming to convert the teacher- led to student-

centred pedagogical practices, such as post-war BiH, the educators' roles in FtFPI call for a reconsideration of how to foster a high-quality FtFPI process to support "success for all" [13]. To accomplish this, the teacher's role and preparation in implementing CL practice require teachers to modify their actions towards FtFPI in responding to pupils' socializing and working together [23,42]. Thus, this study has explored in-depth FtFPI' features for small CL group work to find ways of enhancing pupils to become socially responsive co-learners and cooperative peers.

While implementing FtFPI in CL classrooms does not come without appropriate pupils and teachers' preparation, future studies of FtFPI in CL approach are necessary to accentuate training to promote interpersonal behaviour and supportive communication. Furthermore, using the qualitative and quantitative methodology, a larger sample size would be needed in future studies to examine the variation of socio-pedagogical tools for each aspect of FtFPI.

Ultimately, these findings are encouraging but also limited because only four groups could be videotaped. Moreover, a major limitation of this qualitative study is the reliability factor. However, this study, situated in authentic classrooms, may have some replicable factors for similar studies of peer primary groups using the same data sources and analytical procedures. As the findings here have been confirmed in other studies, they will have practical implications for implementing FtFPI group practice.

**Funding:** This research received no external funding.

**Institutional Review Board Statement:** The study was conducted according to the guidelines of the Declaration of Helsinki, and approved by the Institutional Review Board (or Ethics Committee). Ethical approval for this research was obtained from the Norwegian Social Science Data Service (NSD). The study follows NSD's ethical guidelines, which include securing the confidentiality and anonymity of participants. Project number is 60754.

**Informed Consent Statement:** Informed consent was obtained from all subjects involved in the study.

**Data Availability Statement:** The research data is available at the following link: https://figshare.com/s/f560ec67133266bb99d0, accessed on 21 April 2021.

**Acknowledgments:** My warmest thanks to the participants in this study. Special thanks are extended to Kirsti Klette and Geir Christiansen at the University of Oslo for supporting the realisation of my collected video data. I am very grateful to Tony Burner and Berit Helene Johnsen, Fredrik Mørk Røkenes and NAFOL (The Norwegian National Research School in Teacher Education), PASIE 2 peer group at the University of South-Eastern Norway (Publication Across Subjects in Education) for their helpful feedback on an earlier version of this paper. The author would also like to express her sincere gratitude to the anonymous reviewers and the team of Education Sciences.

**Conflicts of Interest:** The authors declare no conflict of interest.

## Appendix A

**Table A1.** Excerpts of video sessions related to pupils' group assignments.

| Year: 2019 | Code | Excerpt | Videos (min) | School Subject | Task | Purpose |
|---|---|---|---|---|---|---|
| 15 April | SA-G2-Ses.1 | 5<br>7 | 09:35–11:53<br>02:08–03:29 | Mathematics | Different arithmetic operations within the joint task | Preparation for test |
| 16 April | SA-G1-Ses.2 | 9 | 02:40–03:23 | Science | "Plant Detectives" | Analysis of leaves and their structure |
| 17 April | SA-G1-Ses.1 | 8 | 0:23–1:00 | Bosnian Language | Analysis of the main character in the text | Preparation for a debate |
| 19 April | SA-G1-Ses.1<br>SA-G2-Ses.1 | 1<br>1.1<br>6 | 0:24–01:05<br>09:19–09:48<br>03:54–12:07 | Mathematics | Division of a three-digit number by a single-digit number | Exercise |

**Table A1.** *Cont.*

| Year: 2019 | Code | Excerpt | Videos (min) | School Subject | Task | Purpose |
|---|---|---|---|---|---|---|
| 25 April | SB-G2-Ses.2 | 4 | 15:34–16:12 | Mathematics | Multiplication of a three-digit number by a one-digit number | Revision |
| 26 April | SB-G1-Ses.1 SB-G2-Ses.1 SB-G2-Ses.2 | 2 3 3.1 10 | 11:10–11:22 04:46–05:12 13:16–13:40 13:04–13:39 | Mathematics Science | Multiplication of a three-digit number by a one-digit number with transition Past, present and future | Exercise Design mind maps |

## Appendix B

**Table A2.** Clarification of the pupils' FtFPI.

| FtFPI | | Definitions | |
|---|---|---|---|
| Main categories | Sub-categories | Researchers' perspective | Pupils' perspective (Author, 2020) |
| Interpersonal behaviour | Recognising the need for help | Pupils use verbal and nonverbal cues that help them to recognise pupils' signals of confusion (Webb, 1982) Pupils explicitly state about asking for help, Help-seekers persist in asking for help (Webb and Mastergeorge, 2003) | "Pupils' facial expressions show their confusion" "They ask questions or look around" "He would just keep silent" "They are unable to do the task" |
| | Willingness to help | Pupils show their motivation to help one another and facilitate one another's performance with whatever means they have (Slavin, 2015) The help givers expand their efforts to provide relevant help, more elaborated help that is both solicited and unsolicited (Gillies, 2003) | "I first ask her where she is not quite certain" "I ask them whether they need any assistance and if they say yes, I give them an explanation" |
| Supportive communication | Paying attention | Pupils establish eye contact with the speaker and listen actively, e.g., nod, acknowledge the speaker, affirm another pupils' response, make statements that hold the attention of other pupils (Gillies & Ashman,1995) | "Peers look at me and listen, and when I finish they ask me something about what I have been talking about" "They don't interrupt me when I speak" |
| | Encouragement | Making explicit efforts to involve others through verbal and nonverbal gestures; speech or gestures that may encourage the interaction of the group that draws others in (Baines et al., 2009). | "They say something that makes me happy" "I see their smile" |
| | Praising | Promote one another's success that may include eye contact, name use, appropriate statements, pupils' suggestions respected, celebrate success (adapted from Baron, 2003) | "I say super, bravo or you've done this well" "They give me a big hand" |

## Appendix C

Transcription key

Hhhhh audible sigh (exhalation)

.hhh sigh (inhalation)

[. . . ] excluded part of the dialogue

(.) silence, about 1 s

(..) silence, about 2 s

e-e-e words or sounds that are held

! rising intonation

(D.) (saying pupil's first name)

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
