# Peer review of "Becoming a Socially Responsive Co-Learner: Primary School Pupils’ Practices of Face-to-Face Promotive Interaction in Cooperative Learning Groups"

_education, doi:10.3390/educsci11050195_

Round 1
Reviewer 1 Report
Dear authors, I welcome your paper and the attempt to shed light on cooperative learning in pupils of 9-10 years old. Understanding mechanisms that explain cooperative learning is a theoretically interesting and practically important topic.
Besides the strengths of the paper, I provide you several points of feedback to highlight my most critical concerns. As part of this effort, proposed courses of action the author(s) may wish to consider will also be presented.
- mention the data analysis method in the abstract, otherwise, it seems confusing
- also, in the method part, the data analysis method is missing. The coding is followed by a systematic report of the results based on a certain method (content analysis, thematic analysis). Which one have you employed in your analysis?
- my main concern refers to the type of task the children were asked to work on. Even the task is a learning one, sometimes competition between children can emerge. It would be helpful to clarify this aspect
- Also, more information about how the task should be performed is needed. Is it a collaborative task or an individual one?
- Given the children's age, what would be the motivation of the high achiever to help a groupmate?
- What was the teacher's role?
- references 18 and 23 are missing
Author Response
Dear Reviewer 1 of Education Sciences
First, I would like to thank you for your fruitful comments. I am grateful and appreciate your efforts to improve the quality of the manuscript.
Point 1: mention the data analysis method in the abstract,otherwise, it seems confusing
Response 1: The data analysis method has been added to lines 9–11.
Point 2: also, in the method part, the data analysis method is missing. The coding is followed by a systematic report of the results based on a certain method (content analysis, thematic analysis). Which one have you employed in your analysis?
Response 2: Thematic analysis with a hybrid approach that incorporated a deductive, theoretical process and a bottom-up, inductive process was applied. The Data analysis section has been revised in order to clarify this (see lines 224-233).
Point 3: my main concern refers to the type of task the children were asked to work on. Even the task is a learning one, sometimes competition between children can emerge. It would be helpful to clarify this aspect
Response 3: I agree. The importance of the type of task pupils undertake in their groups is a key aspect as it affects their interaction. This aspect is clarified in the text (see lines 103-109).
Point 4: Also, more information about how the task should be performed is needed. Is it a collaborative task or an individual one?
Response 4: It is a collaborative task, but sometimes pupils wrote individually, and later they discussed the solution of the problem as a group. More information is provided in the Data collection section (see lines 191-203).
Point 5: Given the children's age, what would be the motivation of the high achiever to help a groupmate?
Response 6: Very interesting question. Theoretically, linked to the social cohesion perspective (social interdependence theory) holds that pupils help each other, learn because they care about the group and its members and derive self-identity benefits from group membership (Johnson & Johnson, 1998). In essence, the high achievers will help groupmates to learn because they want one another to succeed as indicated by the data in the present study (see lines 110-116 and 619-627). However, the teachers play a crucial role in supporting pupils’ helping each other.
Point 6: What was the teacher's role?
Response 6: The teacher’s role is presented to underline its importance in the revised section 1.1. (see lines 133-138) and Discussion section 1.4. (see lines 586-592).
Point 7: references 18 and 23 are missing
Response 7: After revision of the manuscript, reference 23 is moved to reference 24. Both reference 18 and 24 are blinded because the author’s name is hidden but known to the editor.
Reviewer 2 Report
This is a relevant study for the scientific and educational community but requires certain changes for its acceptance and improvement:
It is not clearly presented what the objective(s) of the study are. Main objective and specific objectives.
It is clear that the article focuses on a case study, but the process followed for the selection of students from each school that makes up the case study should be clearly described. In this way, the study will be better understood.
Could you indicate how the description was made, was any tool used? If so, it would be necessary to indicate and describe it.
Although it is described how to proceed to the data analysis, this section requires an in-depth improvement of the methodological part, it is not clearly presented.
The conclusions should be improved and written based on the objectives of the study. Their presentation and description should be improved, since they denote results but not conclusions of the study.
Although the limitations are presented, prospective studies or future studies should also be presented.
Author Response
Dear Reviewer 2 of Education Sciences
First, I would like to thank you for your fruitful comments. I am grateful and appreciate your efforts to improve the quality of the manuscript.
Response to Reviewer 2 Comments
Point 1: It is not clearly presented what the objective(s) of the study are. Main objective and specific objectives.
Response 1: This should be more clear after revision of the Introduction section (please, see lines 64-71).
Point 2: It is clear that the article focuses on a case study, but the process followed for the selection of students from each school that makes up the case study should be clearly described. In this way, the study will be better understood.
Response 2: Since those participants (N=16) have been selected earlier for the previous study [18], the same pupils were invited to a follow-up video study. Lines 163-178 have been revised in order to clarify this.
Point 3: Could you indicate how the description was made, was any tool used? If so, it would be necessary to indicate and describe it.
Response 3: Although video analysis tools can facilitate the data analysis, a choice for the manual analysis of qualitative data was preferred and described in the Data analysis section in the revised manuscript (see lines 236-240).
Point 4: Although it is described how to proceed to the data analysis, this section requires an in-depth improvement of the methodological part, it is not clearly presented.
Response 4: This should be clearer after the revision of the Data analysis section, lines 224-233, and in-depth insights (lines 262-268).
Point 5: The conclusions should be improved and written based on the objectives of the study. Their presentation and description should be improved since they denote results but not conclusions of the study.
Response 5: The Conclusion section has been revised to make this point clearer and align it with the study's objectives (see lines 656-678).
Point 6: Although the limitations are presented, prospective studies or future studies should also be presented.
Response 6: The future studies of FtFPI in the CL approach are added according to this comment (please, see lines 686-691).
Round 2
Reviewer 1 Report
The authors have revised their manuscript according to my previous observations and integrated the comments appropriately. However, I have small suggestions:
- revise line 50
- typos in line 116
- line 168 - instead of males and females, I suggest using boys and girls
Author Response
Dear Reviewer 1
The author is grateful for your suggestions, and they have been revised accordingly.
Point 1: revise line 50
Response 1: Line 50 has been revised. Instead of “…when pupils participate in cooperative activities”, I revised it to “during cooperative activities.”
Point 2: typos in line 116
Response 2: Done
Point 3: line 168 - instead of males and females, I suggest using boys and girls
Response 3: Revised as suggested. Females and males are replaced with boys and girls.
Reviewer 2 Report
The authors' review has been at least 85% compliant. After reviewing the manuscript, I consider it acceptable for publication.
Author Response
Dear Reviewer 2
The author is grateful for your peer-reviewing.